# Do Genomic Factors Play a Role in Diabetic Retinopathy?

**DOI:** 10.3390/jcm9010216

**Published:** 2020-01-14

**Authors:** Andrea P. Cabrera, Finny Monickaraj, Sampathkumar Rangasamy, Sam Hobbs, Paul McGuire, Arup Das

**Affiliations:** 1Department of Surgery/Ophthalmology, University of New Mexico School of Medicine, Albuquerque, NM 87131, USA; apcabrera@salud.unm.edu (A.P.C.); fmonickaraj@salud.unm.edu (F.M.); sdhobbs@gmail.com (S.H.); 2New Mexico VA Health Care System, Albuquerque, NM 87131, USA; 3Translational and Genomics Research Institute (TGen), Phoenix, AZ 85004, USA; srangasamy@tgen.org; 4Department of Cell Biology & Physiology, UNM, Albuquerque, NM 87131, USA; pmcguire@salud.unm.edu

**Keywords:** diabetic retinopathy, genetics, GWAS, whole exome sequencing, blood-retinal barrier, VEGF

## Abstract

Although there is strong clinical evidence that the control of blood glucose, blood pressure, and lipid level can prevent and slow down the progression of diabetic retinopathy (DR) as shown by landmark clinical trials, it has been shown that these factors only account for 10% of the risk for developing this disease. This suggests that other factors, such as genetics, may play a role in the development and progression of DR. Clinical evidence shows that some diabetics, despite the long duration of their diabetes (25 years or more) do not show any sign of DR or show minimal non-proliferative diabetic retinopathy (NPDR). Similarly, not all diabetics develop proliferative diabetic retinopathy (PDR). So far, linkage analysis, candidate gene studies, and genome-wide association studies (GWAS) have not produced any statistically significant results. We recently initiated a genomics study, the Diabetic Retinopathy Genetics (DRGen) Study, to examine the contribution of rare and common variants in the development of different phenotypes of DR, as well as their responsiveness to anti-VEGF treatment in diabetic macular edema (DME). Our preliminary findings reveal a novel set of genetic variants involved in the angiogenesis and inflammatory pathways that contribute to DR progression or protection. Further investigation of variants can help to develop novel biomarkers and lead to new therapeutic targets in DR.

## 1. Introduction

Diabetic retinopathy (DR) is a microvascular complication of diabetes that involves blood–retinal barrier alteration, inflammation, and neuronal dysfunction [1,2,3]. According to the 2017 International Diabetes Federation Atlas, about 425 million people have diabetes mellitus in the world, and by 2045 this number is projected to reach 629 million [4]. With 35% of the diabetic population afflicted, DR is the most common cause of blindness among middle-aged working adults [5]. Duration of diabetes is the strongest predictor for the progression of DR. Interestingly, some diabetics do not develop DR at all, or only develop mild DR (few microaneurysms), in spite of a long duration of diabetes [6]. Similarly, not all diabetics develop the sight-threatening phenotype of diabetic macular edema (DME) or proliferative diabetic retinopathy (PDR) [7,8,9,10,11]. Furthermore, the response to anti-VEGF (vascular endothelial growth factor) drugs in DME patients is variable, with only 27–45% patients responding well (>15 letters of vision improvement) [12]. The variability in phenotype of DR and anti-VEGF treatment responsiveness in DME suggests a potential role for other factors in the development of DR. 

Several risk factors have been associated with the prevalence of DR. Large-scale epidemiological studies revealed that duration of diabetes, hyperglycemia, hypertension and hyperlipidemia are the major risk factors associated with this disease (see Table 1) [7,13,14,15,16,17,18]. Although there is strong clinical evidence supporting the role of blood glucose, blood pressure, and lipid level in controlling the slow progression of DR (Diabetes Control and Complications Trial [DCCT], United Kingdom Prospective Diabetes Study [UKPDS], Action to Control Cardiovascular Risk in Diabetes [ACCORD]) [14,15,16,17,18,19,20,21] the Wisconsin Epidemiologic Study of Diabetic Retinopathy (WESDR) showed that hemoglobin A1C, cholesterol and blood pressure only account for 10% of the risk for developing retinopathy [22]. Furthermore, a follow-up statistical analysis of the Diabetes Control and Complications Trial (DCCT) revealed that the glycemic exposure (duration of diabetes, HbA1C level) explains only 11% of the decrease in retinopathy risk [23]. Other factors, genetic and/or environmental, may explain the remaining 89% of the variation in retinopathy risk [23]. Together, these observations suggest that genetic factors may play a role in the development and progression of DR. 

Mechanistic studies have identified four major hyperglycemia-induced biochemical pathways associated with DR (polyol, advanced glycation end products, protein kinase C, and hexosamine). These pathways have been shown to lead to detrimental downstream cascading events (oxidative stress, inflammation, and vascular dysfunction) [24]. Despite the success in identifying these biochemical mechanisms and the ability of pharmacological interventions to block these pathways in animal models, these therapeutic strategies have not proven to be efficacious in human clinical trials. Based on these experimental and clinical outcomes, it has become critical to explore alternative factors which may be involved in the development of DR. 

## 2. Role of Genetics in DR

Since the introduction of hereditary transmission by Gregor Mendel, the field of genetics has flourished tremendously [25]. Rapid developments in technology have set the stage for gene mapping and the investigation of numerous disease-associated genetic variants. What started as simple observations in pea plants has evolved into sophisticated methods for investigating the genetic components of complex, multifactorial diseases such as DR [26,27].

Familial clustering studies have consistently shown the involvement of genetics in DR. Early observations of non-insulin-dependent diabetic twins revealed 95% agreement in the degree of severity of this disease [28,29]. Additionally, the DCCT study has shown that diabetic first-degree family members of study subjects who progressed to severe non-proliferative diabetic retinopathy (NPDR) or PDR had a risk ratio of 3.1 for progression compared with those study subjects who did not have such progression [30]. Furthermore, differences in the frequency and severity of DR have long been observed among different ethnic populations [31,32,33]. Together these studies provide evidence in support of the role of genetics in DR. Thus, in this review, we discuss (1) the current understanding of DR genetics and (2) assess recent key studies. Lastly, we propose strategies to address the challenges of previous studies with the goal of furthering insight into the underlying genetic architecture of DR.

Although the role of genetics in DR is well recognized, the precise gene variant(s) underlying this disease remain elusive. While studies have identified many DR-associated genetic variants, only a few have been replicated. However, these confirmatory studies have all resulted in weak associations. Thus, it is likely that these results are indicative of the elaborate disease mechanisms underlying DR. Revisiting previous studies may help in understanding the pitfalls as well developing new strategies to further understand the genetics of DR.

## 3. Heritability and Linkage Analysis

Early studies of sibling pairs have long established the role of genetics in DR. Linkage analyses have provided a foundational method that relies on the physical proximity of non-random associations of alleles of chromosomal mutations to identify disease-associated links [34,35,36]. Thus far, this method has had a long history of success in the identification of variants in monogenic diseases [37,38,39]. However, success in elucidating the role of genetics in complex diseases, including DR, has been arduous [40,41].

To date, cohorts of Pima Indians and Mexican Americans have been studied for DR-associated linkages [42,43,44]. However, these studies have yielded varying results. Interestingly, no common linkage regions were identified in two separate analyses of Pima Indians, despite examination of the same cohort. [42,43]. While one study with Pima Indians demonstrated linkage in the same chromosomal region (1p36) as the Mexican American cohort, the threshold suggestive of linkage by conventional criteria (Logarithm of Odds score > 3.3) was not met. [42,43,44].

While these studies provided strong evidence for genetic contribution in this disease, the lack of reproducibility of the identified DR-associated linkages may be indicative of the involvement of additional factors. The genetic understanding of DR presents a unique challenge because of the etiological mechanisms involved. While DR is recognized as a complex multifactorial disease, understanding disease pathogenesis is further complicated by virtue of retinopathy being a mere, but detrimental, complication of another complex disease (diabetes) [45,46]. To address this challenge, several approaches have been utilized to understand the underlying role of genetics in DR. 

## 4. Candidate Gene Association Studies 

Candidate gene association is an epidemiologic approach frequently used to understand the pathological processes involved in disease [47]. In contrast to gene mapping methods, in which the precise location of genes on the chromosome can be linked to disease, candidate gene association relies on hypothesis-driven inferences with an emphasis on pathological observations [48,49]. Supported, but limited, by clinical observations and biochemical pathway knowledge, this approach provides a practical method for the identification of genetic variants. To date, many candidate genes have been associated with DR: vascular endothelial growth factor (VEGF), hypoxia-inducible factor 1-alpha (HIF1A), and erythropoietin (EPO) genes [50,51,52,53]. Additionally, several glucose metabolism, vascular tone, blood pressure regulation, and inflammatory-associated genes have been identified (receptor for advanced glycation end product (RAGE), aldose reductase (AKR1B1), glucose transporter 1 (SLC2A1), angiotensin-1 converting enzyme (ACE), nitric oxide synthase 3 (NOS3), and intracellular adhesion molecule-1 (ICAM1)) [51,53,54]. However, these studies have yielded variable results. These candidate genes have been previously reviewed elsewhere [55,56]. Importantly, the Candidate gene Association Resource (CARe) study showed that, among 39 genes known to be associated with DR or diabetes, three single nucleotide polymorphisms in P-selectin were associated with DR [57]. None of the genes reported in the candidate gene studies have been replicated in other cohorts. Here, we highlight the vascular endothelial growth factor (VEGF) gene because of its therapeutic success. 

## 5. Vascular Endothelial Growth Factor 

The importance of VEGF in ocular neovascularization was first established in studies of laser-induced hypoxia in non-human primate models [58]. These studies revealed elevated levels of VEGF in aqueous fluid that correlated with the severity of neovascularization. This key finding has since made VEGF a strong proponent in the development of DR [59]. The role of VEGF was further confirmed when clinical observations revealed elevated VEGF levels in the vitreous and aqueous fluid of DR patients [60]. To date, several genetic studies have identified various VEGF polymorphisms associated with DR [50,61,62]. However, these studies have produced variable results [51,57,63].

The putative role of VEGF in ocular neovascularization led to adaptation of anti-VEGF therapies for the treatment of DR [64]. Interestingly, despite the success of anti-VEGF in restoring visual acuity in PDR patients, success has been limited in DR patients with diabetic macular edema (DME), [65] indicating the possible influence of genetic polymorphisms. In a recent study, VEGF polymorphism C634G was identified as a genetic risk factor for DME and its presence resulted in a ‘good response’ outcome to anti-VEGF therapy [66]. However, VEGF polymorphism C634G as a pharmacogenetic marker has yet to be confirmed in follow-up studies. Additionally, it should be noted that VEGF polymorphism C634G has yielded varying results among different population groups [67,68,69]. While the candidate gene association approach has provided valuable genetic and mechanistic insight, strategies that address study variability and lack of reproducibility are yet emerge. Nevertheless, with rapid growth and advancement in the field comes the promise of ever-evolving approaches that can aid in expanding the current understanding of the role of genetics in DR.

## 6. Genome-Wide Association Studies (GWAS)

Genome-wide association study (GWAS) approaches have enabled the identification of hundreds of genetic variants associated with complex diseases by the screening of single nucleotide polymorphisms (SNPs) across the complete genome for disease associations [70]. The first successful GWAS identified disease-associated SNPs in three independent studies of age-related macular degeneration (AMD) [71,72,73]. The success of these studies is commonly attributed to disease heritability, where heritability accounts for 50% of the genetic disease-associated variants in AMD [74,75]. However, this level of genetic heritability is not shared amongst other complex diseases, including DR. For example, Crohn’s disease and Type 2 Diabetes can only be explained by 20% and 6% heritability, respectively [76,77]. Despite the case-to-case variability in heritability, GWAS has proven to be a powerful tool for the identification of SNPs in numerous complex diseases [78].

To date, GWAS has been used to identify DR-associated risk genes in various populations: Texan Mexican-Americans, American Caucasians, Taiwanese, Chinese, Japanese, and Australians (see Table 2) [79,80,81,82,83,84,85]. These studies have been extensively reviewed elsewhere [55,56,86,87]. Recently, GWAS identified genetic variation near the GRB2 gene (downstream of rs9896052, on chromosome 17q25.1) to be associated with sight-threatening DR [85]. To date, these results are the first to be confirmed with reproducible results in independent cohorts. Previously, all DR GWAS have yielded variable results and have lacked reproducibility. One possible explanation for the varying outcomes be the inconsistencies in definitions of DR and controls used in these studies. Additionally, a unique challenge of GWAS is that this method presents a high probability for false positives, due to the vast amount of genetic information yielded from whole-genome mapping. This can be problematic due to genetic variants identified using this method being often located on non-coding genetic regions of the genome [78]. Since it is believed non-coding regions do not offer functional relevance, [74] it is hypothesized that exome-focused approaches may yield a better understanding of genetic associations in DR.

## 7. Whole Exome Sequencing

Whole exome sequencing (WES) methods rely on genome mapping specific to the protein coding (exome) regions [88]. Although exomes comprise only ~1% of the human genome, it has been speculated that exomes harbor ~85% of disease-associated variants [74]. Thus, WES has emerged as a novel and efficient method to identify gene variations that could help explain the role of genetics in complex diseases such as DR.

Recently, the WES approach has been used to identify the genetic variants associated with DR in two independent studies (see Table 3) [89,90]. Shtir and colleagues based their study on an ‘extreme’ phenotype design to search for ‘protective’ gene variants in a Saudi population, hypothesizing that using stringent criteria for study controls would enhance the probability to yield robust candidate variants [89]. Thus, individuals with 10 years duration of diabetes and no sign of retinopathy served as controls, while excluding those with high myopia, advanced glaucoma, and ocular ischemic syndrome, which have been previously shown to offer protection from DR. The DR phenotypes studied were NPDR and PDR with varying severity. Three genes were identified as protectant variants (NME3, LOC728699, and FASTK). 

More recently, Ung and colleagues used a similar approach to analyze an African American (AA) Type 2 diabetic cohort from the African American Proliferative Diabetic Retinopathy Study and a mixed ethnicity (ME) cohort that included Type 1 and Type 2 diabetic participants of African American, Caucasian, and Hispanic backgrounds [90]. The DR phenotype under study was PDR and these cases were compared to the AA Type 2 diabetic control cohort which had a duration of diabetes for a minimum of 10 years. Together, AA and ME cohorts revealed a potential role of 25 novel variants in 19 genes associated with DR. Furthermore, expression-level validation studies demonstrated the potential role of six of the candidate genes identified to play a role in DR pathogenesis. However, one major drawback of this study was the use of the AA cohort as a control for both the AA and ME cohorts.

To our knowledge, these have been the only DR WES studies to date. While both studies revealed novel DR-associated gene variants, these independent studies yielded variable results. The discrepancy of these results may be due to population heterogeneity and varying case definitions for DR phenotypes. However, one major limitation observed in these studies is the definition of controls with regards to no retinopathy despite 10 years of diabetes. As it may take up to 15 years to develop some features of DR, as shown in the WESDR study, controls with no DR should ideally be chosen from patients with a longer duration of diabetes (at least 20 years or longer). Despite the success in the identification of DR-associated variants, these studies must be replicated for meaningful biological conclusions and be further functionally validated. Nevertheless, these studies have provided valuable insight into the role of genetics in DR. 

## 8. Lessons Learned and Road Ahead

At present, the genetic understanding of DR remains convoluted. Identifying the genetic factors responsible for DR has used traditional linkage analysis, candidate gene studies, GWAS, and WES analysis [42,43,44,45,50,51,52,53,54,57,79,80,81,82,83,84,85,89,90]. Out of three linkage studies done in Pima Indians and Mexican Americans, only one study showed a logarithm of odds score of 3.01 for single point and 2.58 for multiple-point analysis at 1p36 in Pima Indians. Additionally, association studies for numerous candidate genes, including VEGF, have yielded variable results [57]. The lack of overall success with the candidate gene studies includes failure to comprehensively identify variation in the genes of interest and incorrect hypotheses about which candidate genes are involved in the disease. Further, GWAS and WES studies for DR have also not produced any genome-wide statistically significant results [79,80,81,82,83,84,85,89,90]. These studies have lacked success due to (1) variability in case definitions that include examination of different DR case definitions including NPDR, PDR, and DME within and between studies, (2) inconsistently defined controls with regards to the duration of diabetes, and (3) population heterogeneity (e.g., discovery and replication samples coming from completely different ethnic populations). With such heterogeneity in phenotype definitions, let alone population heterogeneity, it is not surprising the findings have varied or have not been successful at all. If identified, the genetic factors that contribute to DR can be of added clinical value to determining a person’s risk for DR. Thus, we have recently initiated a genomics study, the Diabetic Retinopathy Genetics (DRGen) Study, in efforts to address previous challenges, further understand the contribution of environmental factors on rare variants, responsiveness to anti-VEGF treatment, and determine if there are variants which protect against the initiation of DR.

## 9. Diabetic Retinopathy Genetics (DRGen) Study Approach

The DRGen Study is a collaboration of UNM School of Medicine and Harvard’s Joslin Diabetes Center. Using a well-defined, clinically supported phenotypic strategy, we seek to better understand the role of rare variants in DR progression, or protection, and anti-VEGF response in DME. We propose for the first time a comprehensive genetic study of the genes and genetic variations involved in the inflammatory and angiogenesis pathways (Figure 1). Using whole exome sequencing (WES) technology, we aim test the coding region of all human genes for associations with DR. In addition to WES, all samples will be genotyped for ancestry informative markers for purposes related to admixture mapping.

Our interest lies in genes known to be involved in inflammatory and angiogenesis pathways, as both processes are known to play a role in DR pathology but have shown weak associations (in heterogeneous sample collections) with DR previously [91,92]. To date, two studies, as described above, utilized the WES technique in extreme DR phenotype patients. However, both studies used a very loose definition of the ‘extreme’ phenotype (i.e., no diabetic retinopathy despite at least 10 years of diabetes) [89,90]. The inclusion of participants with such a short duration of diabetes may lead to the misclassification of controls, given the results of Harvard’s Joslin Medalist Study [7]. Thus, the DRGen study has an emphasis on the phenotypic heterogeneity of DR.

## 10. Phenotypic Heterogeneity in DR

After a period of no DR (clinical absence of vascular lesions in the retina) for a variable period of time (7–10 years), retinopathy develops. DR is classically divided into a non-proliferative (NPDR) or a proliferative (PDR) stage (Figure 2) [6,93,94]. The earliest clinical signs are microaneurysms, followed by dot and blot intraretinal hemorrhages. With the leakage of lipid (hard exudates) and plasma (edema), diabetic macular edema (DME) develops in some NPDR patients. Classically, the natural course of DR is thought to be no DR, mild–moderate NPDR, followed by PDR with longer duration of diabetes. However, DR appears to be a heterogenous disease, in which not every patient will go through the same sequence of events. Furthermore, not all diabetics develop DME or PDR [6]. New vessels grow in a subset of advanced DR patients, resulting in pre-retinal and vitreous hemorrhage, and eventually traction retinal detachment may occur in some patients. The WESDR Study has clearly shown that only about 50% of type 1 diabetics will develop PDR during their life time in spite of long durations of diabetes [8]. Currently, it is not known what protects the other 50% of diabetics from PDR. 

## 11. ‘Extreme’ Phenotype

Furthermore, some diabetics, despite long durations of diabetes (20 years or longer), do not show any sign of DR, or show minimal NPDR (a few microaneurysms). Harvard’s Joslin Medalist Study reported this “extreme phenotype” in about 40% of their diabetics with a duration of 50 years diabetes or longer [7]. In fact, diabetics who did not develop advanced retinopathy over long durations of diabetes are unlikely to experience further worsening of retinopathy once they have had 17 or more years of follow-up. However, it is unknown what factors “protect” these patients from developing DR. Thus, the DRGen study aims to harmonize clinical data to reduce phenotypic heterogeneity; NPDR, PDR, and DME. Additionally, cases and controls will be defined consistently for the different samples studied. Important co-variates such as the duration of diabetes, hemoglobin A1C levels, and ancestry will also be included in the harmonization. Studying extreme phenotypes will further reduce issues related to phenotypic heterogeneity seen in previous studies. 

## 12. DME and PDR: Two Distinct Disease Processes

Interestingly, our preliminary findings revealed that DME and PDR may be two distinctive disease processes [95]. In a retrospective cross-sectional study at UNM, we examined a sample of 165 eyes (majority Hispanics and Native Americans) with a new diagnosis of PDR with active neovascularization and 166 eyes with a new diagnosis of DME. Among the PDR eyes, only 15.7% of eyes (95% CI 9.5–21.8%) had DME by clinical examination or optical coherence tomography measurements of central retinal thickness (Figure 3). Thus, the majority of PDR patients did not have concurrent DME, leading us to ask: why do not all patients with PDR show concurrent vascular leakage (DME) in spite of high VEGF levels? Similarly, among the eyes with DME, only 20.3% of eyes (95% CI 13.5–27.1%) had concurrent PDR. Thus, the majority of DME patients did not have concurrent neovascularization or PDR. Stratified risk factor assessment demonstrated that neither gender, age, type of diabetes, HbA1C, mean arterial pressure (MAP) nor LDL control were statistically significant in the development of DME in the PDR patients, or PDR in DME patients. Therefore, PDR and DME appear to represent two distinct disease processes of the same spectrum, possibly driven by distinctive molecular mediators and possibly distinct genetic factors (Figure 3).

Furthermore, the response to anti-VEGF injections is variable in DME and PDR patients. New retinal vessels in PDR regress completely with one or two anti-VEGF injections in most patients [96], whereas such a robust effect is hardly seen in DME patients [97]. Interestingly, intravitreal anti-VEGF injection is the first line of treatment in DME patients, although the response is suboptimal in many. The variability in treatment responsiveness or differential efficacy of anti-VEGF drugs in DME and PDR suggests that there may be separate molecular pathways and genetic risk factors for the development of DME and PDR. To date, all three major clinical trials (DRCR, RIDE/RISE, VISTA) with anti-VEGF drugs have shown that only 27–45% of DME patients show three-line visual acuity improvement [98,99]. A post hoc analysis of the DRCR Protocol I data revealed that another 30–40% of DME patients do not respond completely to anti-VEGF therapy [12]. Based on our preliminary findings and clinical evidence, we hypothesize that genetic factors may play a significant role in the susceptibility of DR, as well as in the response to anti-VEGF therapeutics. DME and PDR appear to be mediated by separate molecular mediators where inter-individual variation in responsiveness (“good responders” vs. “poor responders”) to anti-VEGF therapy in DME may be attributable, in part, to genetic variants. Thus, this is suggestive that VEGF may be a good pharmacogenetic marker rather than a disease identifying variant.

## 13. Admixture Mapping

The heterogeneity in the populations studied previously may explain the lack of reproducibility of DR genetics studies. It is well known that DR, as with diabetes generally, varies across ethnicities [100,101]. Interestingly, studies of the same ethnic group have also failed to reproduce similar results. Previous studies have relied on self-reported ancestry, which could potentially be an issue if cases and controls are unintentionally pooled from different ethnic groups. If one of the cohorts has higher disease prevalence than the other, one will be overrepresented and the other underrepresented, potentially yielding a high probability of false-positive results [102,103]. Thus, to overcome this limitation, admixture mapping has been used to identify the genetic factors associated with a phenotype in heavily admixed populations [104]. Admixture-based association analyses rely on methods to quantify the degree of ancestry both across the genome as a whole and within defined genomic regions [105]. To overcome this challenge, the DRGen study will use Infinium Multi-Ethnic Global SNP Array, not necessarily to identify individuals by their ancestry, but rather to quantify degrees of admixture, in particular genomic regions that can then be correlated with a phenotype to identify chromosomal regions harboring variants likely to be associated with the DR phenotype [106]. 

## 14. Preliminary Findings

Using the aforementioned study design, two cohorts of patients were selected from the DRGen study population established at the UNM School of Medicine [107]. Briefly, we analyzed an ‘extreme’ phenotype cohort (no DR despite >25 years of diabetes; *n* = 6) and an ‘advanced’ DR cohort (PDR within 15 years of diabetes; *n* = 6). All subjects were matched for gender and age. After obtaining informed consent, DNA was isolated from white blood cells and WES was performed using the SureSelect All Human XT v5 exome kit, analyzed on the Illumina NovaSeq platform, followed by in-house downstream analysis pipeline to align the sequence reads and complete variant calling and annotation. We tested the enrichment of “risk” alleles in cases (PDR Group) with MAF <0.05% and identified four heterozygous missense variants and a frame shift mutation in the PDR group. The analysis of rare coding variants revealed a novel set of genetic variants involved in the angiogenesis and inflammatory pathways that contribute to DR progression (KLF17, ZNF395, CD33, PLEKHG5, and COL18A1) or protection (NKX2.3). These variants are of particular interest, as KLF17 and ZNF395 have been postulated to promote the downstream activation of VEGF [108,109]. Similarly, CD33, a transmembrane receptor expressed on cells of myeloid lineage such as monocytes, has also been suggested to play a role in VEGF expression and inflammation [110]. Furthermore, PLEKHG5 is responsible for encoding a protein leading to NFKB1 signaling pathway activation, known to be involved in DR pathogenesis [111]. Additionally, it is well known that COL18A1 regulates the expression of endostatin, a potent endogenous angiogenesis inhibitor [112]. NKX2.3, a member of the NKX transcription factor family, has been shown to regulate genes involved in immune and inflammatory response, cell proliferation and angiogenesis [113]. While these variants have not been studied well in the context of DR, our preliminary analysis of mRNA isolated from human retinal endothelial cells treated with high glucose, has shown increased expression of COL18A, ZNF395, and PLEKHG5 (*p* < 0.0001). Further validation of these variants is necessary to confirm our findings. At present, the DRGen study is actively enrolling patients with selected DR phenotypes.

## 15. Future Perspectives

Although clinical evidence indicates that genetic factors are implicated in DR, their precise role remains elusive. We recognize that our preliminary findings represent the “tip of the iceberg” and therefore future plans include acquiring a larger study cohort and collecting additional biospecimens (blood and vitreous). Furthermore, we acknowledge that DR is not a homogenous phenotype, thus we will continue to harmonize the clinical data as described herein. The rare variant hypothesis represents the beginning of our vision of a comprehensive strategy involving clinical, genomic, and molecular data coupled with traditional statistical analyses and higher dimensional data analyses (e.g., deep learning). Furthermore, we hope to better understand the role of genetics in the variable anti-VEGF response observed in DME patients. We believe that the approach of harmonization of phenotypes and stringent patient cohort criteria, may lead to identification of novel genetics-based drug targets for DR. We recognize that the value of such results can help lead the forefront in personalized medicine that could potentially diagnose and help choose more efficacious treatments for patients. Immediate plans towards these future directions include creating a repository of blood samples with extracted genetic material and clinical phenotype information as a resource for the research community.

## Figures and Tables

**Figure 1 jcm-09-00216-f001:**
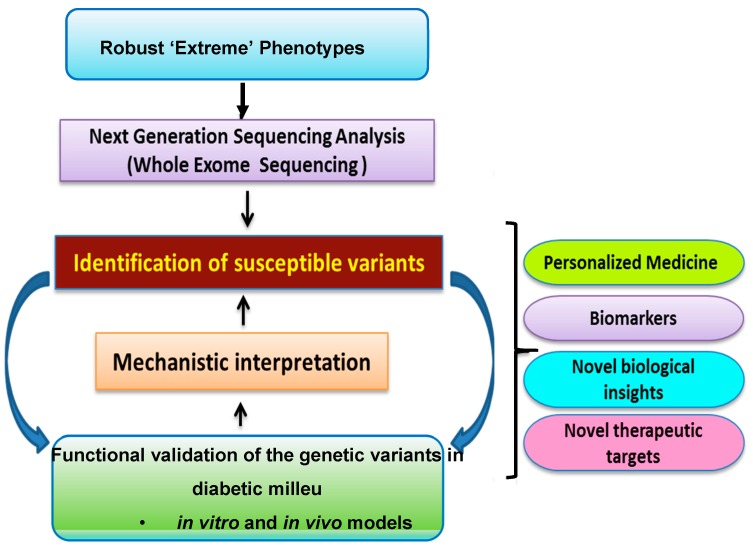
Schematic diagram of the experimental approach for characterizing genetic variants using next-generation sequencing (NGS).

**Figure 2 jcm-09-00216-f002:**
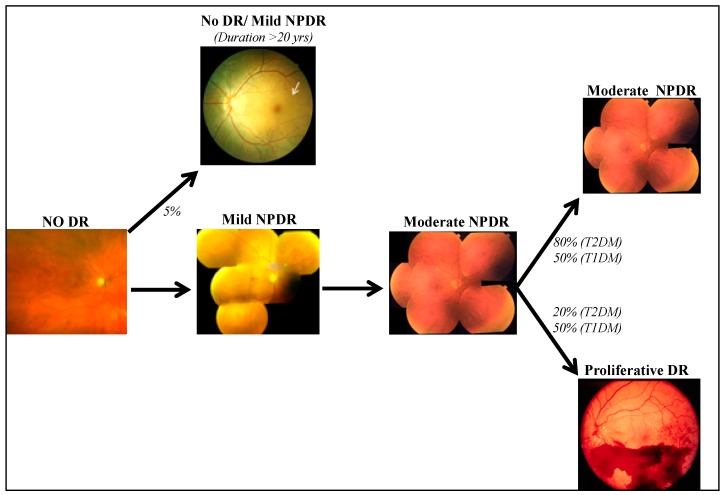
Based on our clinical observations and large epidemiological studies, the hypothesis proposed is that not every diabetic retinopathy (DR) patient goes through the same sequence of events. After a period of diabetes, patients can develop mild non-proliferative diabetic retinopathy (NPDR), followed by moderate NPDR. 20% of type 2 diabetes patients develop proliferative diabetic retinopathy (PDR) while 50% of type 1 diabetes patients develop PDR. In PDR patients, only 15% develop concurrent diabetic macular edema (DME), and the other 85% never develop any macular edema. Interestingly, ~5% diabetic patients never develop DR or only have mild NPDR (1 or 2 microaneurysms, as indicated by white arrow), in spite of 20 or more years of diabetes (“Extreme Phenotype”). Images of mild NPDR and moderate NPDR phenotypes courtesy of the ETDRS Diabetic Retinopathy severity scale *Ophthalmology* (1991).

**Figure 3 jcm-09-00216-f003:**
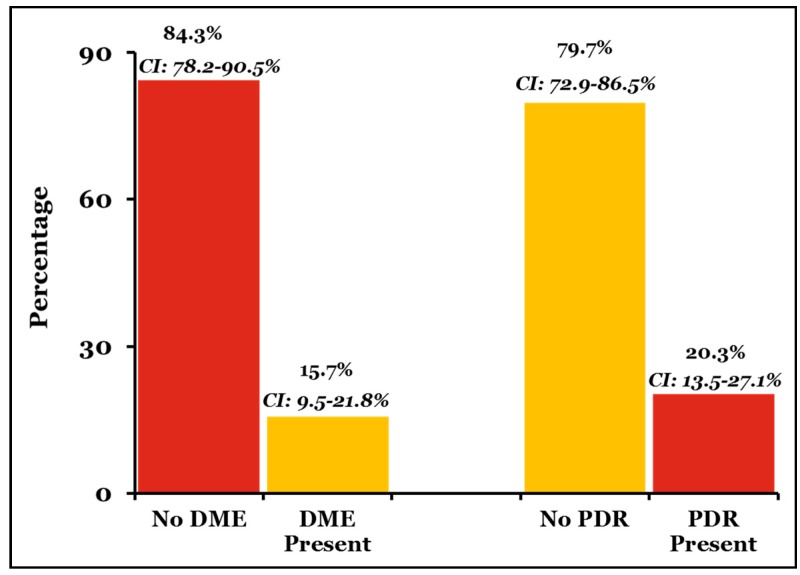
Based on a retrospective clinical study at the University of New Mexico, about 85% of proliferative diabetic retinopathy (PDR) patients do not have concurrent diabetic macular edema (DME), and, similarly, 80% of DME patients do not have concurrent PDR. There is no influence of systemic factor control in such a correlation.

**Table 1 jcm-09-00216-t001:** Risk Factors Associated with Prevalence of Diabetic Retinopathy.

Risk Factor	Study	Major Findings	Mechanism of Action
Duration of Diabetes	50-year Medalist Study	Despite >50 years diabetes duration, no DR observed in ~50% of diabetics	Protection effect from advanced
	(Joslin Diabetes Center) [7]	No association between glycemic control and prevalence of DR	glycation endproduct combinations,
	Wisconsin Epidemiologic Study of	Nearly all type 1 diabetic persons and ~80% of type 2 diabetics develop some	high plasma, carboxyethyl-lysine, and
	Diabetic Retinopathy (WESDR) [8,13]	retinopathy after 20 years of diabetes	pentosidine
Hyperglycemia	WESDR [8,13]	Incidence of Diabetic Macular Edema (DME) over a 10-year associated with higher concentration of glycosylated hemoglobin	
	Diabetes Control and Complications t	Tight glucose control (HbA1c < 6.05%) in Type 1 diabetics prevented development of DR by 76% and slowed progression by 54%	Attributed to increased levels of IGF-1
	Complications Trial (DCCT) [16]	Worseniing of retinopathy in ~10% of DR patients with too tight glucose control	or insulin that can further upregulate VEGF,
		(HbA1c < 6.05%)	resulting in cotton-wool spots and
			blot hemorrhages
	Epidemiology of Diabetes Interventions	10 years after the end of the DCCT study, the benefit of early tight control persisted	∙Histone posttranslational modification by
	and Complications (EDIC) [21]	with risk of retinopathy progression reduced by 53%	acetylation or methylation
	Action to Control Cardiovascular	Type 2 diabetic persons (HbA1c level of 6.4% in intensive group vs. 7.5% in	
	Risk in Diabetes (ACCORD) Eye Study [19]	conventional group) reduced DR progression by 35% over a 4-year span	
		Study discontinued after 3.7 years due to mortality in tight glucose control group	
Hyperlipidemia	Early Treatment Diabetic Retinopathy	DR patients who responded poorly to laser treatment and had diffuse edema with	Agonist action on peroxisome
	Study (ETDRS) [16]	hard exudates had higher levels of blood lipids	proliferator-activated receptor α pathway
	Fenofibrate Intervention and Event	Less need for laser treatment in those treated with lipid lowering drugs	
	Lowering in Diabetes (FIELD) Study [17]	(fenofibrate; 200 mg/day)	
	ACCORD Eye Study [19]	Fenofibrate and Simvastatin cocktail therapy in type 2 diabetics slowed progression	
		of DR at 4 years	
Hypertension	United Kingdom Prospective	Type 2 diabetics showed significant benefit of controlling blood pressure	Angiotensin-converting enzyme inhibitors or b-adrenergic blockers
	Diabetes Study (UKPDS) [14]	(targeting a systolic blood pressure <150 vs. <180 mmHg with standard control)	
	ACCORD Eye Study [19]	No benefit of tight blood pressure control observed	
	Action in Diabetes and Vascular Disease:	No benefit of tight blood pressure control observed	
	Preterax and Diamicron Modified Release		
	Controlled Evaluation (ADVANCE) [18]		

**Table 2 jcm-09-00216-t002:** Genome-Wide Association Studies of Diabetic Retinopathy-associated Risk Genes in Various Populations.

Study	Population	DR phenotype	Control	Identified Variants
Fu et al.	Mexican-American (Texas)	Varying Severity of	No DR-early NPDR; DM 12 ± 9 yrs	CAMK4 and FMN1
J Opthal 2010 [79]	(T2D)	NPDR and PDR (*n* = 103)	(*n* = 183)	
Grassi et al.	American Caucasian from	focal laser treatment for DME	no laser treatment;	rs476141, rs227455, CCDC101
Hum Mol Gen 2011 [80]	GoKinD and EDIC studies (T1D)	panretinal photocoagulation for PDR	DM 24 ± 7 yrs (GoKind), 11 ± 4 yrs (EDIC)	
		(*n* = 973)	(*n* = 1856)	
Huang et al.	Taiwanese (T2D)	NPDR (*n* = 102), PDR (*n* = 72)	No DR; DM 8 ± 6 yrs	PLXDC2, ARHGAP22
Ophthalmology 2011 [81]			(*n* = 575)	
Sheu et al.	Taiwanese from Taiwan-US	PDR (*n* = 437)	No DR; DM ≥ 8 yrs	TBC1D4-COMMD6-UCHL3,
Hum Mol Gen 2013 [82]	Diabetic Retinopathy (TUDR)			LRP2-BBS5, ARL4C-SH3BP4
	Study (T2D)			
Lin et al.	Taiwanese (T2D)	Varying Severity of	No DR; DM 5–10 yrs	rs10499298, rs10499299, rs17827966,
Ophthalmologica 2013 [83]		NPDR and PDR (*n* = 174)	(*n* = 575)	rs1224329, rs1150790, rs713050,
				rs2518344 and rs487083; all associated with
				genes TMEM217, MRPL14 and GRIK2
Awata et al.	Japanese (T2D)	Varying Severity of	No DR; DM 7 ± 6 yrs	rs9362054
PLoS One 2014 [84]		NPDR and PDR (*n* = 837)	(*n* = 1149)	
Burdon et al.	Australian (T2D)	Sight-thretening DR	No DR; DM ≥ 5 yrs	rs3805931,
Diabetologia 2015 [85]		NPDR and PDR (*n* = 336)	(*n* = 508)	rs9896052 (down stream of GRB2 gene)

**Table 3 jcm-09-00216-t003:** Whole Exome Sequencing Studies of Diabetic Retinopathy.

Study	Population	DR phenotype	Control	Identified Variants
Shtir et al.	Saudi	Varying Severity of	No DR, DM 10 yrs	NME3, LOC728699, FASTK
Hum Genet 2016 [89]	(T1D and T2D)	NPDR and PDR (*n* = 43)	(*n* = 64)	
Ung et al.		PDR (*n* = 57)	No DR, DM 10 yrs (*n* = 13)	
Vis Res 2017 [90]				
	African American			AKR1C3, KIAA1751, CD96, CRIPAK, RGMA,
	(T2D)			ZNF77, MPZL3, NLRP12, FAM92A1, EFCAB3,
				HNRNPCL1, SIGLEC11, ATP12A, TMEM217,
				FAM132A, SLC5A9
	Mixed Ethnicity			ABCA7, ABHD17A, ANO2, BPIFB6, C15orf32,
	(T1D and T2D)			CCDC105, CDKL1, CEP192, COL6A5, CRIPAK,
				DNHD1, GPATCH1, HMCN1, KIF24, LRBA, LRB8,
				MSH2, NAT1, PHF21A, PKHD1L1, SLC6A13,
				SLURP1, TTC22, UPK3A, VPS13B, ZDHHC11B,
				ZDHHC11, ZNF600

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
