# Peer review of "Do Genomic Factors Play a Role in Diabetic Retinopathy?"

_jcm, 2020, doi:10.3390/jcm9010216_

Round 1
Reviewer 1 Report
The authors have given good details on DR and genetics. However, some details on Genomic factors in line with gene therapy for diabetic retinopathy will also help the readers, especially the current therapy development using adeno-associated virus (AAV), CRISPR, stem cell, etc. If there is word limitation, It can be also another separate review article in future.
Author Response
Reviewer #1
The authors have given good details on DR and genetics. However, some details on Genomic factors in line with gene therapy for diabetic retinopathy will also help the readers, especially the current therapy development using adeno-associated virus (AAV), CRISPR, stem cell, etc. If there is word limitation, it can be also another separate review article in future.
We thank the Reviewer for the comment. We agree with the Reviewer that genomic factors in line with gene therapy for diabetic retinopathy would be beneficial to the readers. We are aware of the ongoing studies in line to develop new therapeutics using adeno-associated virus (AAV), CRISPR, and stem cells. Interestingly, transduction using AAV vector of endostatin, a cleavage product of one of the variants in our advanced proliferative diabetic retinopathy cohort (COL18), has already been found to reduce retinal neovascularization in mice (Biswal et al., 2014). While gene therapy is in line with the genetic studies presented in this review, we believe these topics are out of the scope of this manuscript. We plan on writing an extensive review in the future.
Reviewer 2 Report
Thank you for allowing me to review your paper. It is well written and a good review paper. It serves as a good introduction to the work you are doing which will be useful going forward in the understanding and management of diabetic retinopathy.
Only suggestion, more detailed explanation of the work done in Section 13 to be added in the revision. The abstract will need a minor revision to reflect this change.
Author Response
Reviewer #2
Thank you for allowing me to review your paper. It is well written and a good review paper. It serves as a good introduction to the work you are doing which will be useful going forward in the understanding and management of diabetic retinopathy. Only suggestion, more detailed explanation of the work done in Section 13 to be added in the revision. The abstract will need a minor revision to reflect this change.
We thank the Reviewer for the comment. As suggested, we have included more details on our preliminary findings (Section13). This change is also reflected in the revised abstract.